# Fluorogenic RNA Mango aptamers for imaging small non-coding RNAs in mammalian cells

Alexis Autour[1], Sunny C. Y. Jeng[2], Adam D. Cawte [3,4], Amir Abdolahzadeh[2], Angela Galli[2], Shanker S.S. Panchapakesan[2], David Rueda [3,4], Michael Ryckelynck[1] & Peter J. Unrau [2]

Despite having many key roles in cellular biology, directly imaging biologically important RNAs has been hindered by a lack of fluorescent tools equivalent to the fluorescent proteins available to study cellular proteins. Ideal RNA labelling systems must preserve biological function, have photophysical properties similar to existing fluorescent proteins, and be compatible with established live and fixed cell protein labelling strategies. Here, we report a microfluidics-based selection of three new high-affinity RNA Mango fluorogenic aptamers. Two of these are as bright or brighter than enhanced GFP when bound to TO1-Biotin. Furthermore, we show that the new Mangos can accurately image the subcellular localization of three small non-coding RNAs (5S, U6, and a box C/D scaRNA) in fixed and live mammalian cells. These new aptamers have many potential applications to study RNA function and dynamics both in vitro and in mammalian cells.

[1] Université de Strasbourg, CNRS, Architecture et Réactivité de l'ARN, UPR 9002, 67000 Strasbourg, France. [2] Department of Molecular Biology and Biochemistry, Simon Fraser University, 8888 University Drive, Burnaby, BC V5A 1S6, Canada. [3] Single Molecule Imaging Group, MRC London Institute of Medical Sciences, Du Cane Road, London W12 0NN, UK. [4] Department of Medicine, Molecular Virology, Imperial College London, Du Cane Road, London W12 0NN, UK. Alexis Autour, Sunny C. Y. Jeng and Adam D. Cawte contributed equally to this work. Correspondence and requests for materials should be addressed to D.R. (email: david.rueda@imperial.ac.uk) or to M.R. (email: m.ryckelynck@unistra.fr) or to P.J.U. (email: punrau@sfu.ca)

Since their introduction, fluorogenic RNA aptamers that enhance the fluorescence of an unbound fluorophore have sparked significant interest and hold great potential to enable the visualization of RNA molecules within a cell[1–4]. However, developing high contrast aptamer-fluorophore systems with brightness comparable to existing fluorescent proteins has posed a significant experimental challenge. In an ideal system, unbound fluorophores with high extinction coefficients and low quantum yields become highly fluorescent when bound by an RNA aptamer whose tertiary structure correctly positions the fluorophore into an orientation that maximizes its brightness[1,5–7]. While reported aptamer-fluorophore complexes make use of fluorophores with high extinction coefficients, notably RNA Mango[8] and the cytotoxic Malachite Green binding aptamer[5], these systems suffer from low quantum yields. Conversely, systems with high quantum yields such as the GFP-mimic aptamers[1,2,9,10] have intrinsically low extinction coefficients. As a consequence, such complexes are all significantly less bright than enhanced GFP[11], diminishing their utility

High-affinity aptamers, with the notable exception of RNA Mango, have also been difficult to develop. While not important for a perfect fluorophore with zero unbound quantum yield, high-affinity fluorophore aptamer complexes allow lower free fluorophore concentrations to be used during imaging, effectively decreasing background fluorescent signal[12]. Despite the inability to simultaneously optimize aptamer-fluorophore brightness and binding affinity, existing fluorogenic systems have achieved some notable successes in bacteria, yeast and mammalian cells[1,2,13–15]. This suggests that using newly developed screening methodologies to select brighter fluorogenic RNA aptamers either by FACS[9] or droplet-based microfluidics platforms[10] can provide powerful and easy to use fluorescent RNA imaging tags to study cellular RNAs.

Here, we have used a competitive ligand binding microfluidic selection to isolate three new aptamers (Mango II, III and IV) with markedly improved fluorescent properties, binding affinities, and salt dependencies compared to the original Mango I aptamer[8]. These aptamers all contain a closing RNA stem, which isolates a small fluorophore-binding core from external sequence, making them easy to insert into arbitrary biological RNA. Unexpectedly several of these constructs are resistant to formaldehyde, allowing their use in live-cell imaging and also in conventional fixed cell methodologies. Stepwise photobleaching in fixed cell images indicate that as few as 4–17 molecules can be detected in each foci, and photobleaching rates in live cells or in vitro were at least an order of magnitude slower than found for Broccoli. These new aptamers work well with existing fluorescence microscopy techniques and we demonstrate their applicability by imaging the correct localization of 5S, U6 and the box C/D scaRNA (mgU2-47) in fixed and live mammalian cells. Together, these findings indicate that the new Mango aptamers offer an interesting alternative to existing fluorogenic aptamers[12].

## Results

**Microfluidic isolation of new and brighter Mango aptamers.** Mango I is an RNA aptamer that was initially selected from a high diversity random sequence library for its TO1-Biotin (TO1-B) binding affinity rather than for its fluorescent properties, which may have precluded the enrichment of the brightest aptamers in the library[8]. Its structure consists of a three-tiered G-quadruplex with mixed parallel and anti-parallel connectivity (Fig. 1)[16]. The observation that the RNA Spinach aptamer can form a 4.5-fold brighter complex with TO1-B than Mango I, in spite of its significantly lower affinity[17], also suggests that more fluorogenic Mango-like folds may exist in the library. To address

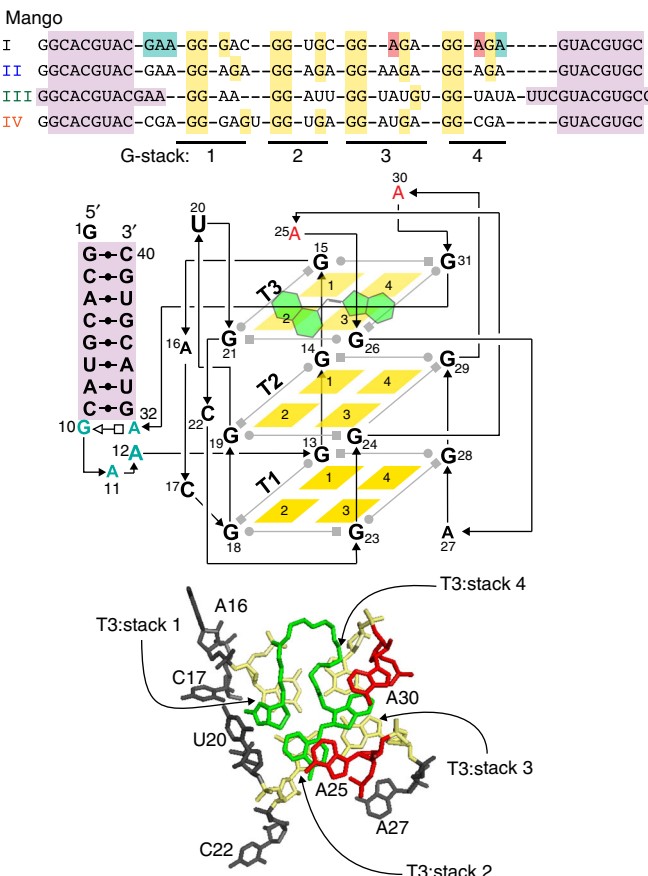

**Fig. 1** RNA Mango aptamers core sequences. Colour-coded alignment of RNA Mango I, II, III and IV. G residues in yellow are protected from dimethyl sulfate (DMS) cleavage when folded in the presence of fluorophore. Quadruplex stacks and their associated propeller sequences are numbered 1 through 4. The GAAA isolation motif of Mango I, together with two adenines essential for binding, are shown in green and red respectively. Purple shading represents a flanking stem region for all four Mango aptamers. Schematic: a tertiary structure of Mango I, showing tier 1, 2 and 3 of its quadruplex structure (T1, T2 and T3) and colour-coded as in top panel. TO1-B is shown in green. Bottom: top view of the Mango I core (PDB ID: 5V3F[16]), showing the T3 tier of the quadruplex and relevant propeller residues, colour coding matches the schematic and top panel

this, we rescreened the original round 12 Mango I library (R12) using microfluidic-assisted in vitro compartmentalisation (μIVC, Fig. 2a)[10]. Interestingly, the initial screening shows that a significant fraction of molecules in the R12 library are brighter than Mango I (Fig. 2b).

A potential limitation of μIVC is the requirement of high TO1-B concentrations (~100 nM, due to the requirement for high speed fluorescent sorting) during the screening. Such high values would greatly exceed the Mango I $K_D$ (~3 nM) and could potentially prevent the selection of high-affinity aptamers. To mitigate this, we supplemented the in vitro transcription (IVT) mixture with TO1-B competitors NMM (N-methyl mesoporphyrin IX)[18] and TO3-Biotin[8] (Supplementary Fig. 1), which are both known to interact with G-quadruplexes. As expected, the NMM supplemented IVT mixture significantly reduces TO1-B/Mango I fluorescence (Supplementary Fig. 2). The NMM concentration was progressively increased during the first four screening rounds, therefore, any brightness increase at each round presumably results from the selection of brighter aptamers in the library, while retaining high affinity and selectivity for

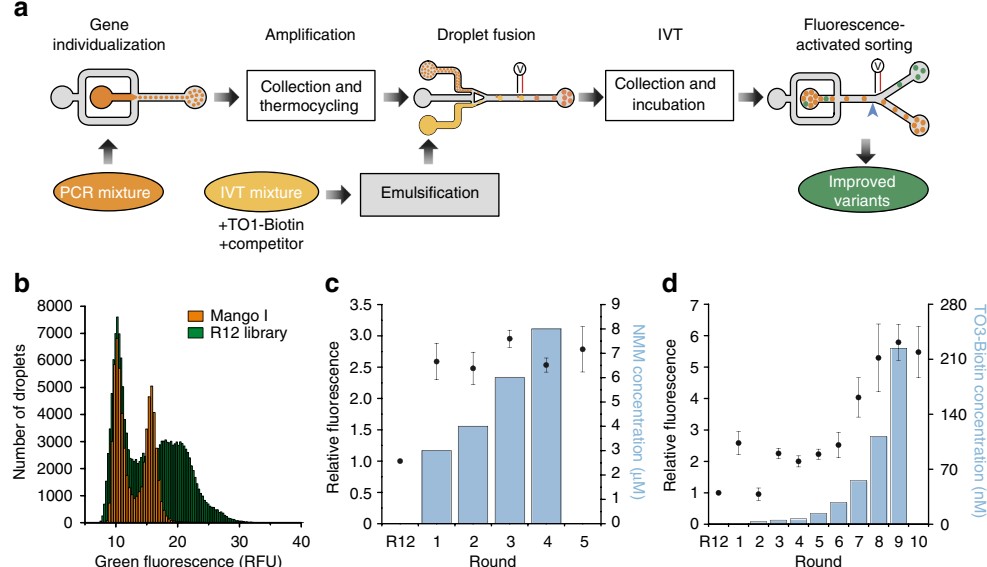

**Fig. 2** Competitive selection of TO1-B-binding variants using droplet-based microfluidics fluorescence screening. **a** Experimental workflow for microfluidic-assisted fluorescence screening. Ovals and boxes represent on- and off-chip steps, respectively. Three microfluidic devices were used for gene individualization in 2.5 pL droplets containing PCR mixture; after thermocycling, fusing each PCR droplet with a droplet containing an in vitro transcription (IVT) mixture supplemented with TO1-B and competitor (NMM or TO3-Biotin); and, after incubation, analysing the fluorescence profile of each droplet and sorting them accordingly. **b** Fluorescence profile of droplets containing Mango I or the initial R12 library (~200,000 variants, Supplementary Table 1). Droplets containing no DNA have a fluorescence of 10 RFUs. **c** Improvement in fluorescence enhancement of aptamer libraries during the screening process in the presence of increasing amounts of NMM. The fluorescence (black dots) of the RNA libraries in complex with TO1-B was determined by mixing 2 μM RNA and 100 nM TO1-B in the absence of NMM. These values were normalized to that of the starting library (R12). The values are the mean of three independent experiments and error bars correspond to ±1 standard error. **d** Enhancement in fluorescence resulting from selection with TO3-Biotin competitor. The fluorescence (black circles) was determined after each round by mixing 300 nM RNA and 100 nM TO1-B in the absence of TO3-Biotin. The values were normalized to that of the starting library (R12). The blue bars represent the concentration of competitor used in each round of selection. The values are the mean of three independent experiments and error bars correspond to ±1 standard error, and for each sort, the gated populations can be found in Supplementary Fig. 3

TO1-B (Fig. 2c). The stability of the RNA/TO1-B complex was further challenged by sorting the droplets at 45 °C as previously described[10]. The relative fluorescence of the library increased 2.5-fold in the first round (~3 million variants analysed, Supplementary Table 2), but it did not increase further over the later rounds (Fig. 2c, Supplementary Fig. 3a). However, the ability of NMM to compete against TO1-B binding decreased progressively with each round (Supplementary Fig. 4a), indicating that the aptamers in the later rounds have higher affinity and/or selectivity for TO1-B. The last screening round shows that, in the absence of NMM, the fluorogenic properties of the enriched library remained unchanged (Fig. 2c). From the final enriched library, we cloned and sequenced 32 pool RNAs, and analysed their fluorogenic capacity (Supplementary Fig. 5a, b). While the brightest clone was R5-NMM-20, 6 of the 13 brightest aptamers exhibited an almost identical sequence to clone R5-NMM-5 (Supplementary Fig. 5c).

In a second set of screenings, we increased the selection stringency by using the Mango I specific competitor TO3-Biotin, which differs from TO1-B by having two additional carbons in the methine bridge of TO1-B (Supplementary Fig. 1). To further increase the selection pressure for TO1-B binding, we also decreased the RNA concentration in the droplets to 0.3 μM (from 8 μM with NMM). TO3-Biotin competitor was introduced in the second round of screening to ensure that positive droplets were not missed in the first round (Supplementary Fig. 3b). In subsequent rounds, TO3-Biotin concentration was gradually increased (Fig. 2d and Supplementary Table 3). While the relative fluorescence of the population increased in the first screening round, it decreased upon addition of competitor in round two, likely due to the elimination of brighter but weaker binding

aptamers (Fig. 2d). In later rounds, the relative fluorescence increased progressively until the TO3-Biotin concentration exceeded TO1-B by 2.2-fold (220 nM and 100 nM, respectively). Beyond this ratio, the competition was too high and the selection process collapsed. The final round shows that the enriched library maintains its fluorescent properties in the absence of competitor. RNA molecules from each of the final rounds were cloned and sequenced. Interestingly, further characterization of the different libraries indicates that aptamers were first selected for their capacity to discriminate TO1-B from TO3-Biotin (Supplementary Fig. 4b, c). Surprisingly, we found that, starting from round three, the libraries were dominated by a single cluster of sequences (cluster E, Supplementary Fig. 6) that was attributed to the TO3-resistant aptamers discussed above. In the last four rounds, this sequence was progressively replaced by the point mutant C66U, best represented by the aptamer R10-17 and identical in sequence to R5-NMM-20 found in the NMM competitive screen. The progressive domination by R10-17 was likely the origin of the fluorescence improvement observed in the last rounds of selection. Finally, among the remaining clusters identified in the early rounds, cluster D, represented by R2-1, was found to have particularly high binding affinity (Supplementary Fig. 7a).

**Each new Mango variant is unique in structure and function.** Based on the parental sequence isolates R2-1, R5-NMM-20 (R10-17), and R5-NMM-5 (Supplementary Figs. 5, 6 and 7), we engineered the minimal reference constructs Mango II, Mango III and Mango IV (Fig. 1), respectively, by truncation and sequence manipulation while maintaining the binding and fluorescent

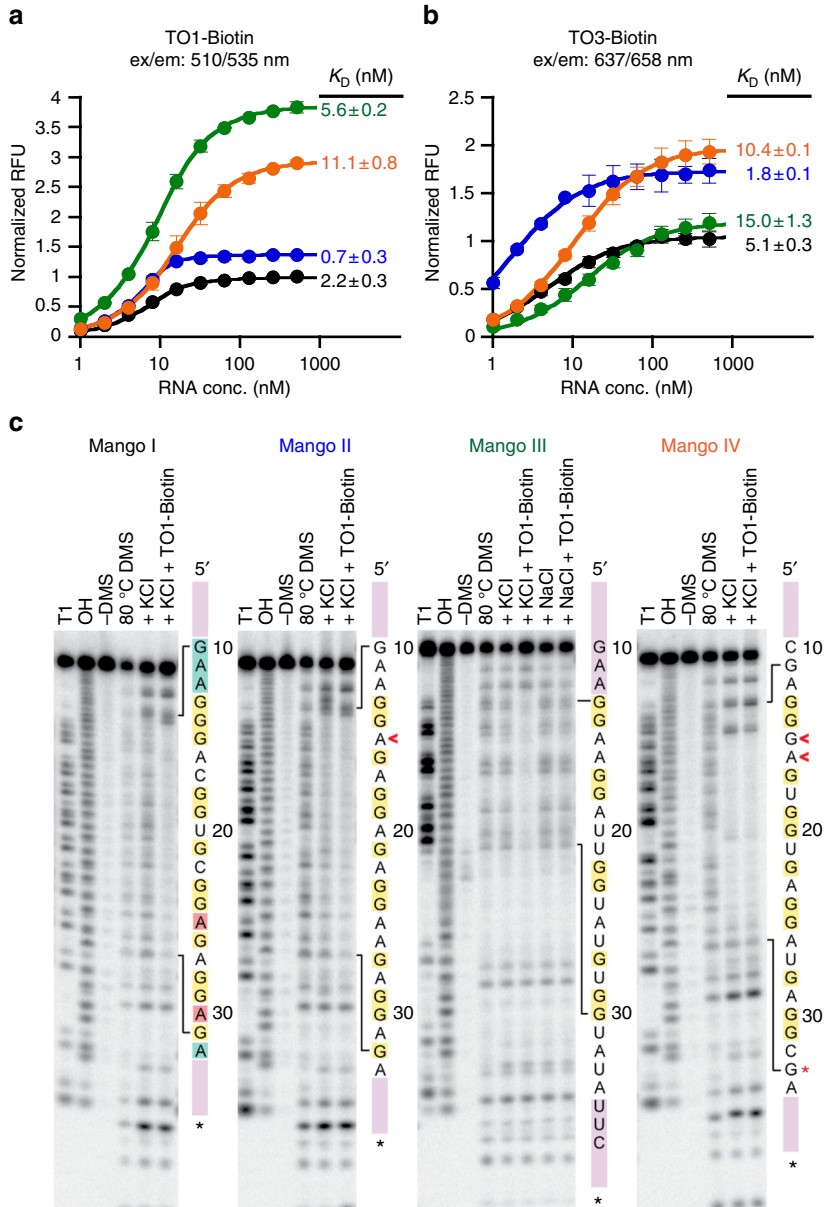

**Fig. 3** RNA Mango aptamers fluorophore-binding and DMS protection. **a** Fluorescence binding curves for each Mango aptamer determined by titrating RNA aptamer concentration while holding TO1-B fluorophore constant at 10 nM. $K_D$ values are shown next to each titration. **b** Same as for **a** but using 1.4 nM TO3-Biotin. Data for **a** and **b** have been normalized such that Mango I has a maximum RFU of 1 in each case. Error bars are standard deviation of three replicates. **c** DMS chemical protection patterns for the four Mango aptamers. 3' end-labelled RNA ($^{32}$P pCp shown as a black asterisk) was subjected to DMS chemical modification followed by reduction by NaBH$_4$ and aniline cleavage as described in the methods. RNA sequences are displayed to the right of each set of lanes with stem portions represented as purple blocks. Legend: T1—denatured T1 ladder; OH—partial alkaline hydrolysis ladder; (−) DMS— denatured reaction with ddH$_2$O added in place of DMS; 80 °C DMS—denatured DMS ladder; remaining lanes are native DMS reactions with addition of potassium to 140 mM final (+KCl), addition of sodium to 140 mM final (+NaCl), with or without 500 nM TO1-B (+TO1-B). Red asterisk indicates a notably unprotected G in Mango IV. Red daggers in Mango II and Mango IV indicate nucleotides presumed to be looped out in the T3 layer relative to the Mango I G stack shown in Fig. 1

properties of the parental constructs (Supplementary Fig. 7). Mango II, III and IV were found to be 1.5-, 4- and 3-fold brighter than Mango I, respectively (Fig. 3a). Mango II binds TO1-B with subnanomolar affinity, while Mango III and IV had slightly weaker affinities than Mango I (Fig. 3a). Mango II and IV also demonstrated improved fluorescence response when bound to TO3-Biotin relative to Mango I while exhibiting nanomolar binding affinities to this strongly red-shifted fluorophore (Fig. 3b). Notably, the brightness of the Mango III and Mango IV TO1-B bound complexes are 43,000 M$^{-1}$ cm$^{-1}$ and 32,000 M$^{-1}$

cm$^{-1}$, respectively, making Mango III 1.3 times brighter than enhanced GFP (EGFP) a common benchmark for the characterization of improved fluorescent proteins[11].

Mango I binds TO1-B by sandwiching it between the T3 layer of the G-quadruplex and A25 and A30 (Fig. 1, yellow and red residues, respectively)[16]. This fluorophore-binding core is isolated from an arbitrary RNA duplex (Fig. 1, purple residues) by a GAAA tetraloop-like adaptor[19] that inserts the TO1-B binding core between the third and fourth residues of the tetranucleotide motif (Fig. 1, cyan residues)[16]. Like Mango I, all of the new

Mango aptamers are contained within an arbitrary closing stem (Supplementary Fig. 7). Dimethyl sulfate (DMS) probing, which correctly confirmed the three-tiered quadruplex structure of Mango I, indicates that Mango II also contains a three-tiered quadruplex structure (Fig. 3c). In distinct contrast to Mango I, which has a contiguous run of protected three guanine residues forming G-stack 1 (Fig. 1, G13-G15 of Mango I), Mango II has an A15 insertion between G14 and G16. Since G13, G14 and G16 are DMS-protected in Mango II, this implies a structural rearrangement of the T3 level of the aptamer (Fig. 3c, '<' symbol). In addition to this change, Mango II has a dinucleotide adenine in its third propeller loop region (Fig. 3c), whereas Mango I has a single adenine in this location. In Mango I, this adenine is stacked on top of the methylquinoline heterocycle of the TO1-B (Mango I A25, Fig. 1 and Fig. 3c, first red residue) implicating an additional structural change in Mango II relative to that of Mango I. Indeed, either of these changes, individually or together, were shown to play an important role in the improved affinity and brightness of Mango II (Supplementary Fig. 7a).

Mango IV has a different fold than Mango I or II and is considerably brighter than either. Unexpectedly Mango IV, which like Mango I contains three contiguous G residues (G13-G15), lacked N-7 protection of residue G15, but like Mango II has strong DMS protection of G17 after A16 (Fig. 3c, '<' symbols). This implies that the dinucleotide G15 and A16 of Mango IV must be in a distinct conformation relative to either Mango I or II. Interestingly, Mango IV also lacks DMS protection of residue G33 (Fig. 3c, * symbol), which in Mango I plays an instrumental role in forming the T3 layer and that is DMS-protected in Mango II. In addition, the GAAA linker motif of Mango I (Fig. 1, blue residues), which is also apparently found in Mango II, does not appear to be present in Mango IV, as replacing the 5′ CGA sequence of the Mango IV core sequence with GAA resulted in a four-fold decrease in binding affinity (Supplementary Fig. 7b, variant 20). These data, together with additional point mutational analysis, indicate that, while Mango IV appears likely to contain T1 and T2 tiers of guanine tetraplexes in common with Mango I and Mango II, its T3 tier is likely to differ considerably in structure from either that of Mango I or II.

The folding of the Mango II and IV constructs was characterized further by examining their salt dependence, thermal melting properties, and CD spectra. Both Mango II and IV have Hill coefficients and affinities for potassium similar to Mango I (Supplementary Table 4 and Supplementary Fig. 8). Mango II showed a limited fluorescence response in the presence of sodium ions, while Mango I and Mango IV did not fluoresce appreciably with this ion. Most notably, in the presence of potassium, these aptamers were resistant to $Mg^{2+}$ levels >10-fold higher than the selection concentration (22 mM), whereas Mango I fluorescence was strongly inhibited at such high concentrations (Supplementary Fig. 8). This indicates that Mango II and Mango IV are substantially more stably folded than Mango I, likely as a result of being selected for fluorescence at elevated temperature and in the presence of high levels of free magnesium. While Mango I and II both displayed a change in DMS protection upon addition of TO1-B to aptamers pre-incubated in potassium buffer, the DMS protection pattern of Mango IV was largely unchanged upon addition of TO1-B (Fig. 3c). Mango II, had thermal melting properties that were largely unchanged whether or not TO1-B was present, while Mango IV exhibited hysteresis in the unbound melting curve. Both thermal melts were considerably different from that of Mango I, which changes its $A_{260}$ thermal melt profile depending on the presence or absence of TO1-B[17] (Supplementary Fig. 9). Consistent with the formation of a G-quadruplex structure in Mango I, II and IV, the ligand bound CD spectra for each aptamer were quite similar (Supplementary Fig. 10).

Mango III, the brightest of the three constructs, contains only nine guanines in its core, and is therefore unable to form a three-tiered G-quadruplex. All nine core guanines are DMS-protected (Fig. 3c). Mutant analysis suggests that the helical region of Mango III is likely to extend an additional 3 bp into its core (Figs. 1 and 3c, light purple), as changing this putative 3 bp duplex had only a modest impact on binding affinity. In contrast, removing either the 5′ GAA or 3′ UUC sequence completely ablated binding (Supplementary Fig. 7c, variants 9, 10 and 11). Consistent with the hypothesis that the C66U mutation observed during selection played a role in higher fluorescence, reverting this mutation in the truncated Mango III context reduced fluorescence by 40% (Supplementary Fig. 7c, variant 12), suggesting that this nucleotide plays an important role in conferring fluorescence. Mango III contains much longer A/U rich propeller regions than any of the other Mangos (Fig. 1) and has a ~100-fold higher affinity for potassium and sodium, while being only modestly inhibited by high levels of magnesium (Supplementary Fig. 8). Its sigmoidal fluorescent melting curve resembles the melting of RNA Spinach2[2] and not the more linear melting curves observed for Mango I, II and IV (Supplementary Fig. 9). Similarly, the CD spectrum of the bound Mango III complex is different in the 270–300 nm region from the other Mango constructs tested (Supplementary Fig. 10). Correlated with this distinct CD spectra, Mango III lacks an excitation shoulder found to be in common for all the other Mango constructs in the 270–300 nm region (Supplementary Fig. 11). This and other differences in the excitation and emission spectra all suggest that Mango III binds TO1-B differently than Mango I, II and IV aptamers. Detailed X-ray structure analysis will be required to uncover further details of this interesting aptamer.

**Cellular imaging of Mango-tagged RNAs**. To test the newly isolated aptamers in cells, we tagged the small and well-characterized human 5S ribosomal RNA with each Mango variant by incorporating an F30 folding scaffold (Supplementary Fig. 12a) previously shown to improve cellular fluorescence and RNA stability[1,20]. Each Mango-tagged RNA, with or without the folding scaffold or terminator hairpin, exhibited comparable fluorescence intensities in vitro (Supplementary Fig. 12b). No appreciable fluorescence was observed in the absence of the TO1-B fluorophore or with the control constructs that contained either the F30 folding scaffold alone, or a G-quadruplex mutant Mango sequence (Supplementary Fig. 12b and Supplementary Table 5).

To image the tagged RNA, we transfected in vitro-transcribed 5S-F30-Mango RNA into HEK293T cells, fixed the cells on ice and stained with TO1-B (Online Methods). This protocol is based on the observation that, in vitro, Mango I, II and IV-fluorophore complexes are substantially resistant to formaldehyde at room temperature (Supplementary Fig. 13a). Up to ~10 bright RNA Mango foci could be readily detected per cell with a fluorescence microscope, but not in control transfections (Fig. 4a). A time course of this process (Supplementary Fig. 14a) shows the initial delivery of lipofectamine particles to the cell membrane (5 min after transfection) followed by dispersal of the RNA in the cytoplasm (15–30 min) and foci formation (30–60 min), indicating that the observed foci are not intact lipofectamine-RNA particles that remain after transfection. Furthermore, these foci cannot correspond to 5S-F30-Mango RNA in late endosomes, given that Mango fluorescence decreases significantly at low pH (Supplementary Fig. 13b). Contrary to 5S-F30-Mango I and III, transfections with 5S-F30-Mango II and IV RNA consistently exhibit visible foci. A possible explanation is that Mango II and IV fold correctly both in the presence and absence of TO1-B, unlike the other Mangos (Supplementary Fig. 9). Consistent with

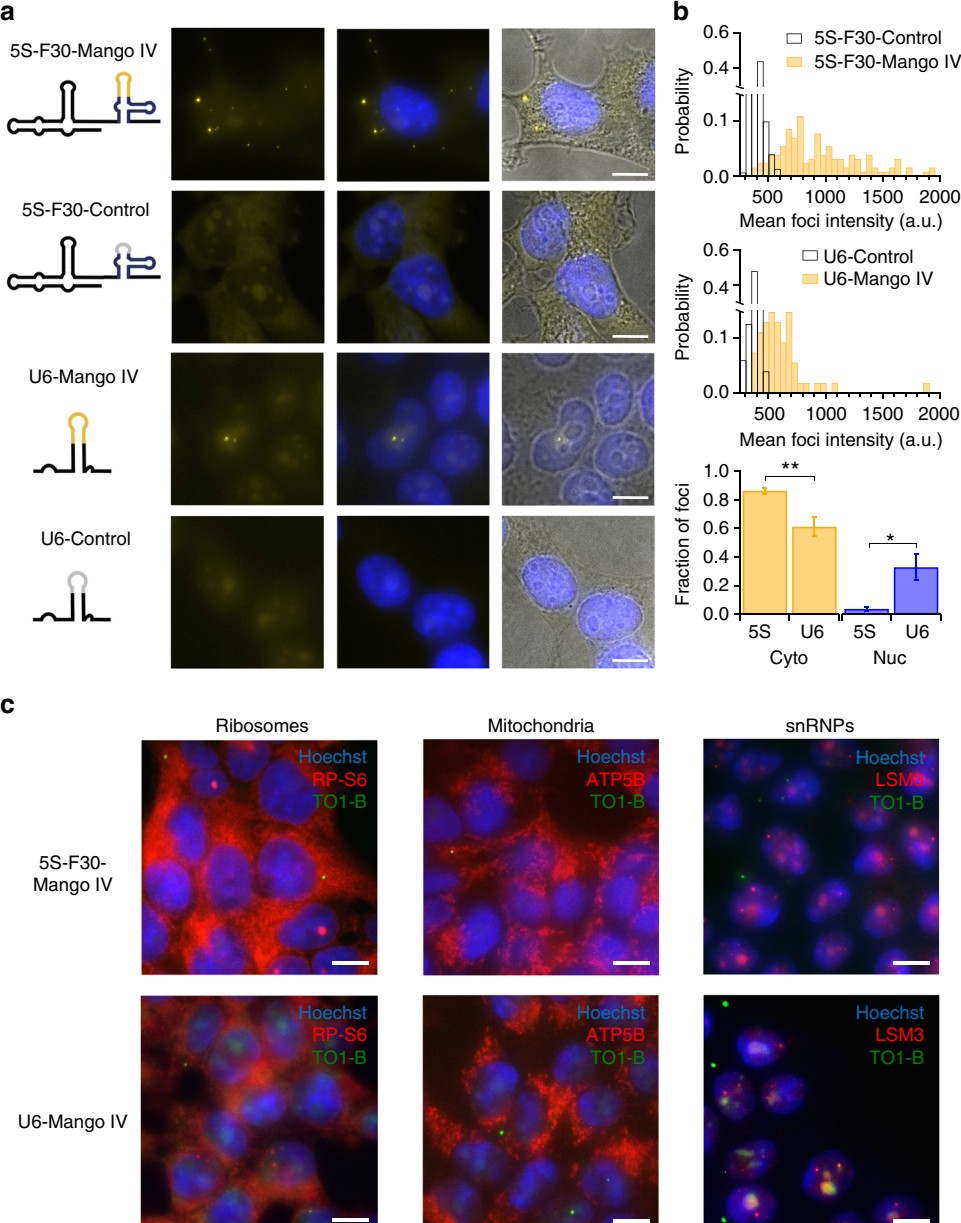

**Fig. 4** Cellular imaging of Mango IV tagged RNAs. **a** Maximum projections of fixed cells containing Mango IV tagged 5S and U6 RNAs stained with 200 nM TO1-B (yellow) and 1 μg/mL Hoechst 33258 (blue)—construct diagrams shown as RNA-Mango (yellow) or non-fluorescent control RNA (grey), F30 folding scaffold (blue) and remaining RNA sequence (black). **b** Mean intensity distributions of 5S-Mango IV and U6-Mango IV foci (yellow) compared to low intensity foci detected in control experiments (black). Fraction of foci observed in the cytoplasm and nucleus for 5S and U6-Mango IV RNAs (bottom panel, *$p < 0.05$ and **$p < 0.01$ calculated using a $t$ test). The number of cells for 5S-F30-Control, 5S-F30-Mango IV, U6-Control and U6-Mango IV were 57, 114, 131 and 183, respectively. Error bars depict standard error in the mean. **c** Localization of 5S-Mango IV and U6-Mango IV relative to immunostained ribosomes (RP-S6), mitochondria (ATP-5B) and snRNPs (LSm3). Scale bars are 10 μm. All images are maximum projections except in **c**, which show a single focal plane

this, similar levels of DMS protection in dye bound and unbound samples were most clearly seen in Mango IV and in part for Mango II (Fig. 3c). The mean intensity of the 5S-F30-Mango IV foci was two- to three-fold higher than 5S-F30-Control background (Fig. 4b). The majority of 5S-F30-Mango IV foci (~85%) are cytoplasmic, a small fraction (~5%) are clearly nuclear and the remaining foci appeared on the nuclear boundary.

To precisely determine the subcellular localization of the 5S-F30-Mango IV foci, we combined Mango-based imaging with immunostaining, which is made feasible by the ability of Mango IV to withstand formaldehyde fixation. As expected, cytoplasmic 5S-Mango IV foci overlap with antibody staining against the ribosomal protein S6 (RP-S6, Fig. 4c). In addition, we observe that the cytoplasmic 5S-Mango IV foci overlap with immunostained mitochondria (ATP5B, Fig. 4c and Supplementary Fig. 15a), and not with constructs lacking Helix IV (Supplementary Fig. 17c, e), consistent with the observation that Helix IV of 5S ribosomal RNA (rRNA) mediates its import into mitochondria[21]. Conversely, we do not observe overlap with other subcellular compartments, such as P-bodies, endosomes or stress granules, where the transfected RNA could be processed for degradation (Supplementary Fig. 15b).

To confirm that the observed foci are specific, we tagged and transfected an RNA that localizes to a different cellular

compartment. The U6 snRNA (small nuclear RNA) is expected to associate with snRNP (ribonuclear protein) complexes in the nucleus. We tagged U6 snRNA by incorporating Mango IV directly into an internal stem loop (Supplementary Fig. 12), known to be amenable to modification[22–24]. The resulting construct exhibits comparable fluorescence intensity to Mango IV alone in vitro (Supplementary Fig. 12, Supplementary Table 5). Direct transfection of U6-Mango IV snRNA yields fluorescent foci comparable to 5S-F30-Mango IV (Fig. 4a), albeit with lower intensity (Fig. 4b). As expected, the fraction of nuclear foci increased approximately ninefold, while cytoplasmic foci decrease significantly ($p < 0.01$, Fig. 4b). As opposed to 5S-F30-Mango IV, cytoplasmic U6-Mango IV foci did not significantly overlap with mitochondria or ribosomes, whereas nuclear U6-Mango IV foci do overlap with snRNP protein LSm3 (Fig. 4c), as expected. Similar to 5S-F30-Mango IV, U6-Mango IV foci do not overlap with other subcellular compartments, such as P-bodies, endosomes or stress granules (Supplementary Fig. 15b). To further quantify the proportion of nuclear and cytoplasmic Mango-tagged RNA, we created two-dimensional (2D) intensity plots for each pixel from multiple images (Supplementary Fig. 16). Intensity thresholds were set above the observed profile of control RNAs and the apparent nuclear boundary. These plots highlight the preferential cellular location of Mango-tagged 5S and U6 RNAs, with 5S predominantly cytoplasmic (~64%) and U6 predominantly nuclear (~73%). We performed a similar analysis to correlate the normalized immunostaining and Mango signals by setting appropriate thresholds above the signal observed in each control (Supplementary Fig. 17a, b). Co-localization with the mitochondrial marker ATP5B is only observed with 5S-F30-Mango IV and not with U6-Mango IV or

the helix IV mutant 5S-Δ78-98-F30-Mango IV shown to be deficient in mitochondrial import[21] (Supplementary Fig. 17c–e). Also, consistent with the images shown in Fig. 4, a high level (90%) of co-localization between U6-tagged RNAs and the snRNP marker LSm3 is observed compared to the U6-Control and 5S-F30-Mango IV RNAs (Supplementary Fig. 17f–i). Taken together, these fixed cell data show that Mango IV can be used to label and image small cellular RNAs via direct transfection of in vitro-transcribed RNAs, without affecting their expected subcellular localization.

To test whether Mango-tagged RNA molecules can be imaged in live cells, we took advantage of the aptamer's high affinity for TO1-B, and transfected in vitro-transcribed 5S-F30-Mango RNAs pre-incubated with TO1-B. After transfection, cells exhibit bright foci only in the presence of each 5S-F30-Mango RNA due to the pre-incubation with TO1-B stabilizing efficient fluorescence (Supplementary Fig. 14b and Supplementary Movies 1 and 2). The foci observed were similar to those in Fig. 4a, albeit with a lower background fluorescence in the nucleolus of fixed cells. The foci can be readily tracked revealing three distinct diffusive behaviours (fast, slow and static) and their respective root mean square displacement coefficients could be quantified (Supplementary Fig. 14c and Supplementary Movie 3). Interestingly, the Mango-based aptamers stably fluoresce under constant or pulsed illumination. Photobleaching curves of constantly illuminated live cells containing 5S-F30-Mango I, IV and dBroccoli show >10-fold improvement in photostability compared with dBroccoli (Supplementary Fig. 14d). These data are in good agreement with in vitro photobleaching analysis in aqueous droplets (Supplementary Fig. 18a). Under pulsed illumination (200 ms, 0.2 Hz), the lifetime of Mango RNA aptamers in fixed cells increases by

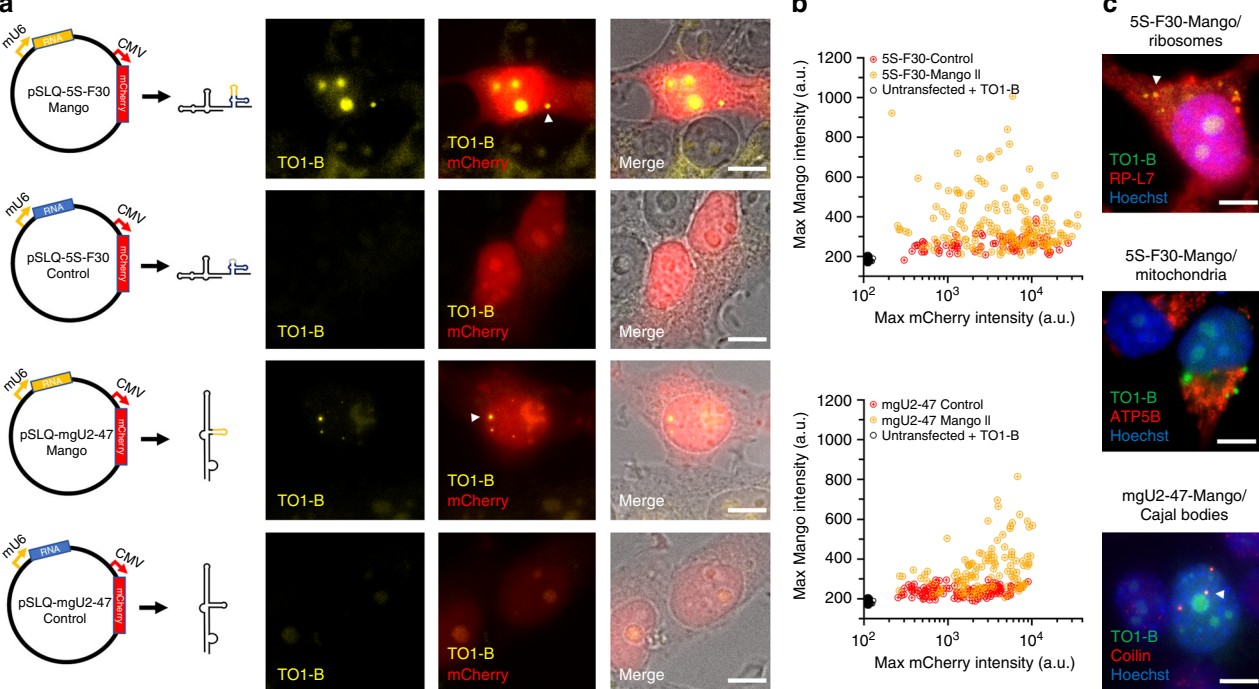

**Fig. 5** Cellular imaging of genetically encoded Mango II-tagged RNAs. **a** Diagram of plasmid constructs with the 5S rRNAs and mgU2-47 scaRNAs under the control of a murine U6 promoter (Pol III) and co-expression of a mCherry reporter gene (CMV promoter). Shown adjacent are images of individual slices of fixed cells either expressing Mango II-tagged RNAs (top) or control RNAs (bottom) with the TO1-B (200 nM) signal in yellow, mCherry in red and brightfield image in greyscale. Arrows depict significant cellular and nuclear foci. Scale bar = 10 μm. **b** 2D maximum intensity plots of individual nucleoli and Mango II specific foci for both the TO1-B signal (y axis) and mCherry signal (x axis—$\log_{10}$ scale). The number of cells for 5S-F30-Control, 5S-F30-Mango II, untransfected cells + TO1-B, mgU2-47 Control and mgU2-47 Mango II were 89, 167, 98, 130 and 117 respectively. **c** Maximum projections of cytoplasmic 5S-F30-Mango IV foci and nuclear mgU2-47 foci from plasmid expression in conjunction with immunostained ribosomes (RP-L7), mitochondria (ATP5B) and Cajal bodies (Coilin). Arrows depict significantly co-localized foci, scale bar = 10 μm

>60-fold (from 11.7 s to >10 min, Supplementary Fig. 18b), as previously observed for the Spinach aptamer[25].

To estimate the number of fluorescent 5S-F30-Mango IV molecules in each foci, we performed photobleaching-assisted microscopy on fixed cells (Supplementary Fig. 19a). A maximum likelihood estimate analysis of the photobleaching trajectories[26,27] revealed between 4 and 17 photobleaching steps per foci. In addition, the photobleaching step distribution reveals two peaks corresponding to either one or two molecules (Supplementary Fig. 19b). The number of observed steps correlates linearly with the initial foci intensity below 2000 intensity units (Supplementary Fig. 19c). Altogether, these results indicate that each foci contains at least 4–17 fluorescent molecules, consistent with the observed range of experimental intensities, and raises the interesting possibility of imaging single molecules in live cells via the incorporation of a small number of Mango repeats.

Finally, to test whether the new Mangos have the ability to function as genetically encoded tags expressed in cells, we constructed plasmids that express the 5S rRNA under the control of a RNA pol III promoter in conjunction with an mCherry reporter gene to identify successfully transfected cells (Fig. 5a). Upon fixation, we observed that cells expressing the pSLQ-5S-F30-Mango II and IV constructs exhibit an increased fluorescent signal in nucleolar compartments as well as forming distinct cytoplasmic foci when compared with the pSLQ-5S-F30-Control construct (Fig. 5a, Supplementary Fig. 20a). The analysis of the peak Mango and mCherry intensities for multiple cells expressing the pSLQ-5S-F30-Mango II plasmid shows a population of cells with a high Mango specific signal, not seen in cells expressing the pSLQ-5S-F30-Control plasmid (Fig. 5b). Interestingly, we observe that cells exhibiting lower mCherry intensities can also show higher Mango signal, consistent with RNA transcription preceding mCherry translation. In agreement with this, reducing plasmid expression time, from 24 to 12 h, increased the number of observed cytoplasmic foci (Supplementary Fig. 20a). Under the same conditions of fixation and staining, signal was not observed in untransfected cells or in cells expressing the 5S-F30-Broccoli construct (Supplementary Fig. 20b). The robust cytoplasmic signal observed after 12 h of pSLQ-5S-Mango IV expression enabled us to combine Mango imaging with immunofluorescence (Fig. 5c). As expected, the observed Mango foci co-localize significantly with Ribosomal Protein L7. However, no significant co-localization was observed with the mitochondrial stain ATP5B. The absence of co-mitochondrial localization, in this case, is likely due to the fact that most nucleolar expressed 5S rRNA will assemble into ribosomes in the nucleus, whereas 5S rRNA molecules transfected directly in the cytoplasm will not, and are more readily available for mitochondrial import. The observed cytoplasmic foci did not co-localize with immunostaining for stress granules, P-bodies or endosomes (Supplementary Fig. 20c).

To confirm that the observed 5S rRNA foci are specific, we expressed a Mango II-tagged small Cajal-body-specific RNA (mgU2-47) that mediates the 2′-O-methylation of the U2 snRNA[28]. Upon expression, the Mango-tagged mgU2-47 RNA formed well-defined nuclear foci that were absent in the mgU2-47 Control RNA (Fig. 5a, b). The nuclear foci also co-localized with immunostained Cajal bodies (Fig. 5c). Taken together, these results demonstrate the ability of Mango tags to function as efficient genetically encoded reporters of RNA subcellular location.

## Discussion

To be of broad utility, fluorescent RNA aptamers should be bright, bind their ligands with high affinity, and be compatible with existing methodologies to image proteins in live and fixed cells. To achieve this, we have developed a novel, competition-based, ultrahigh-throughput fluorescent screening approach that takes advantage of microfluidic-assisted in vitro compartmentalization to select three new and highly effective RNA Mango fluorogenic aptamers. The broad range of novel photophysical and biochemical properties in the new Mango aptamers promises to make them highly competitive with existing aptamer-fluorophore systems[1,5,8,10,29,30].

The new Mango aptamers when bound to TO1-B are very bright. The original Mango I construct when bound to TO1-B exhibits a quantum yield of ~0.14. This quantum yield is similar to that typically observed when thiazole orange is rigidified by intercalation into a duplex nucleic acid[17,31]. Mango III and IV, when bound to TO1-B, have quantum yields of ~0.56 and 0.42, respectively. Such high quantum yields are comparable or exceed that of TO1-activating proteins[32]. When combined with the high absorbance of thiazole orange, these quantum yields result in fluorescence as bright or brighter than EGFP, a significant milestone for aptamer-fluorophore systems.

While a brighter aptamer system has been recently discovered[29], the high brightness of the Mango aptamers combined with their extremely high binding affinity toward TO1-B and the very low unbound fluorescence of TO1-B strongly enables high contrast imaging. The nanomolar binding affinity of these aptamers allows very low concentrations of fluorophore ligand to be used (50-fold less than typically used by the GFP-mimic aptamers). Indeed, transfecting RNA bound to stoichiometric amounts of TO1-B resulted in fluorescent RNA foci that could be tracked in cells, implicating that fluorophore-bound RNA Mango complexes are both stable and robustly fluorescent in living cells, even when at very low effective concentrations. Likewise, staining fixed cells with low concentrations of TO1-B (200 nM) readily generated Mango-tagged fluorescence with only modest levels of nonspecific nucleolar staining being observed, strongly suggesting that nonspecific fluorescence induced by intercalation of TO1-B into nonspecific nucleic acids can be effectively controlled using the Mango system. As modifying thiazole orange to TO1-B considerably destabilizes the weak intercalation of thiazole orange into dsDNA[31] and RNA[8,17], further optimization of the imaging approaches presented here appears likely to enable higher contrast RNA Mango imaging in the future.

Equally important for RNA imaging is the photostability and biological compatibility of the Mango system. The RNA Mango aptamers are at least an order of magnitude more photostable than the Broccoli systems and comparable to the recently published Corn aptamer, albeit Corn's requirement for dimerization seems less compatible with its simple biological utilization[30,33]. Furthermore, pulsed illumination dramatically enhances the imaging time possible with RNA Mango, opening up the interesting possibility to track biological RNAs for 10 min or longer. Just as critical, our results with three small non-coding RNAs demonstrate that Mango tags can be incorporated either into non-essential stems or as 3′-tags, without significantly interfering with RNA subcellular localization and we do not believe a folding scaffold is required for the Mango system as evidenced by the success of the U6 and scaRNA constructs. Indeed, using either genetically encoded Mango-tagged RNA or direct RNA transfections correctly recapitulates the expected localization patterns.

There are many other applications of the Mango systems that appear likely based on the established biochemistry and photophysics of these aptamer complexes. First, as we have demonstrated, via the observation of quantized photobleaching of RNA foci, as few as four RNA Mango can be robustly imaged within one puncta. In the future, tagging biological RNAs with a small number of Mango repeats could, therefore, enable robust single-

molecule RNA imaging. Second, our ability to sort the latest Mango constructs with microfluidics, together with our observation of Mango I dependent fluorescence in bacteria via FACs[8] strongly suggests, at least for highly expressed biological RNAs, that the RNA Mango system can be used to enable RNA-based FACs experiments. Third, the high affinity of Mango I-based tags to TO1-desthiobiotin has been used to recover native RNP complexes from streptavidin beads[34]. Such pulldowns should be readily applicable with our new Mango constructs potentially enabling a unique combination of RNA imaging and RNP pull-down experiments via the insertion of a single Mango tag into a biological RNA of interest. Finally and while not utilized here, the broad salt tolerance of the new Mango aptamers in contrast to that of Mango I and other fluorophore aptamer systems, makes the Mango system compatible with a range of enzymatic reactions. This implies that in addition to the in-cell demonstrations given here, in the future, in vitro fluorescent applications can be developed that make use of the Mango systems high brightness and fluorophore-binding properties.

While the full biological compatibility of the Mango aptamers is still not completely explored, their small size relative to other fluorogenic RNA aptamers, their ability to fold correctly into monomers at physiological temperatures[16], combined with their unusual ability to withstand formaldehyde fixation all promise to be very useful for investigating biological systems in the future.

## Methods

**High-throughput screening.** High-throughput screening proceeds in three major stages:

(i) Digital droplet PCR: DNA libraries were diluted in 200 µg/mL yeast total RNA solution (Ambion) as described before[35] to have a final average number of DNA molecule per droplet ($\lambda$) of ~0.2–1 (Supplementary Tables 1–3). One microlitre of this solution was introduced in 100 µL of a PCR mixture containing 0.2 µM of forward primer (5′−CTTTAATACGACTCACTATAGGGAACCCGCAAGCCATC), 0.2 mM of reverse primer (5′−CAGAATCTCACACAGCC), 0.2 mM of each dNTP, 0.67 mg/mL Dextran-Texas Red 70 kDa (Molecular Probes), 0.1% Pluronic F68, 2 µL Phire II DNA polymerase (Thermo Scientific, concentration unavailable) and the supplied buffer (proprietary to Thermo Fisher) to recommended concentrations. The mixture was loaded in a length of polytetrafluoroethylene (PTFE) tubing (I.D. 0.75 mm tubing; Thermo Scientific) and infused into a droplet generator microfluidic device[10] where it was dispersed into 2.5 pL droplets (production rate of ~12,000 droplets per s) carried by HFE 7500 fluorinated oil (3 M) supplemented with 3% of a fluorosurfactant[35]. Droplet production frequency was monitored and used to determine droplet volume by adjusting pumps flow rates (MFCS, Fluigent). Emulsions were collected in 0.2 µL tubes as described[35] and subjected to an initial denaturation step of 2 min at 95 °C followed by 30 PCR cycles of: 30 s at 95 °C, 30 s at 55 °C, 1 min 30 s at 72 °C.

(ii) Droplet fusion: PCR droplets were then injected into a fusion device[35] at a rate of ~1500 droplets per s, spaced by a stream of HFE 7500 fluorinated oil supplemented with 2% fluorosurfactant. Each PCR droplet was synchronized with a 16 pL IVT droplet containing 2.2 mM of each NTP (Larova), 24 mM MgCl2, 44 mM Tris-HCl pH 8.0, 50 mM KCl, 5 mM DTT, 1 mM Spermidine, 35 µg/mL of Dextran-Texas Red 70 kDa (Molecular Probes), 0.1% Pluronic F68, 3500 U T7 RNA polymerase (purified in the laboratory and estimated to have an activity around 2500 U/µL by comparing it with commercial enzyme), 100 nM TO1-B[8], 5 ng/µL inorganic pyrophosphatase (Roche) supplemented with the desired concentration of NMM. For the screenings performed in the presence of TO3-Biotin[8], the T7 RNA polymerase (New England Biolabs) concentration was reduced to 70 U per reaction. The IVT mixture was loaded in a length of PTFE tubing (I.D. 0.75 mm tubing; Thermo Scientific) that was kept on ice during all the experiment. IVT droplets of 16 pL were produced at a rate of ~1500 droplets per s and paired to one PCR droplet. Pairwise droplets were then fused by electrocoalescence while passing between a pair of electrodes subjected to an AC electric field of 400 V (30 kHz) via high-voltage amplifier (Model 623b, Trek)[10]. The resulting emulsion was collected off-chip and incubated for 120 min (high concentration of T7 RNA polymerase, NMM screenings) or 30 min (low concentration of T7 RNA polymerase, TO3-Biotin screenings) at 37 °C.

(iii) Droplet analysis and sorting: the emulsion was finally re-injected into an analysis and sorting microfluidic device mounted on a Thermo plate (Tokai Hit) holding the temperature at 45 °C as previously described[10]. Droplets were re-injected at a frequency of ~200 droplets per s, spaced with a stream of surfactant-free HFE 7500 fluorinated oil. The green fluorescence (TO1-B in complex with the aptamer) of each droplet was analysed. Between 1 and 2% green fluorescence droplets were gated for each round of selection. The gated droplets were deflected into a collecting channel by applying a 1 ms AC field (1200 V, 30 kHz) and were collected into a 1.5 mL tube. Collected droplets were recovered by flushing 200 µL of surfactant-free HFE 7500 fluorinated oil (3 M) through the tubing. 1H, 1H, 2H, 2H-perfluoro-1-octanol of 100 µL (Sigma-Aldrich) and 200 µL of 200 µg/mL yeast total RNA solution (Ambion) were then added, the droplets broken by vortexing the mixture and DNA-containing aqueous phase was recovered.

**Quantification of RNA produced in droplets.** A PCR mixture supplemented with DNA coding for RNA Mango (starting with 10 copies of template DNA molecules per droplet to ensure that all the droplets were occupied) was emulsified in 2.5 pL droplets and the DNA amplified as above. The droplets were paired and fused with droplets of IVT mixture containing either a low (70 U of enzyme from New England Biolabs) or a high (20 µg/mL of enzyme purified in the lab) concentration of T7 RNA polymerase and the resulting emulsions were incubated for respectively 30 min or 120 min at 37 °C. After incubation, the RNA-containing phase was recovered using 1H, 1H, 2H, 2H-perfluoro-1-octanol (Sigma-Aldrich) and the transcription was stopped by a phenol extraction followed by an ethanol precipitation in the presence of 300 mM sodium acetate pH 5.5 (Sigma-Aldrich). After centrifugation and a wash in 70% ethanol, the pellets were re-suspended in water. Baseline-Zero™ DNase of 10 U (Epicentre) and the corresponding buffer were added and a second incubation of 60 min at 37 °C was performed. The DNase was removed by phenol extraction and RNA recovered by ethanol precipitation.

Recovered RNAs were reverse-transcribed for 60 min at 55 °C, followed by 5 min at 95 °C, in a mixture containing 1 µM of reverse primer, 0.5 mM of each dNTP, 8 U/µL RT Maxima (Thermo Scientific) and the supplied buffer according to recommended concentrations. The complementary DNA (cDNA) was amplified using SsoFast™ Evagreen supermix (Bio-Rad) supplemented with 0.2 µM of each primer (forward and reverse) using a CFX96 Touch™ real-time PCR detection system (Bio-Rad). Finally, the cDNA was quantified using the calibration curve obtained with reactions performed with purified Mango II RNA.

**Enrichment measurement.** The pool molecules contained in 2 µL recovered from the sorted fractions were introduced into 100 µL of PCR mixture containing 0.1 µM of each primer (fwd and rev), 0.2 mM of each dNTP, 0.05 U/µL of DreamTaq™ and its corresponding buffer (Fermentas). The mixture was then subjected to an initial denaturation step of 30 s at 95 °C, followed by 20 cycles of: 5 s at 95 °C and 30 s at 60 °C. 20 µL of PCR products were then in vitro-transcribed in 250 µL of mixture containing 2 mM of each NTP, 25 mM MgCl2, 40 mM Tris-HCl pH 8.0, 5 mM DTT, 1 mM Spermidine and 70 µg/mL T7 RNA polymerase. After 4 h of incubation at 37 °C, 10 U of Baseline-Zero™ DNase (Epicentre) and the corresponding buffer were added and the mixture was incubated for 60 min at 37 °C. RNAs were recovered by phenol extraction followed by an ethanol precipitation in the presence of 300 mM sodium acetate pH 5.5 (Sigma-Aldrich). After centrifugation and a wash in 70% ethanol, the pellets were dissolved in denaturing loading buffer (0.05% bromophenol blue, 0.05% xylene cyanol, 20% glycerol, 1× TBE, 8 M urea) and the solution loaded onto a 12% denaturing 8 M urea polyacrylamide gel. The piece of gel containing RNA was identified by ultraviolet (UV) shadowing, sliced from the gel and transferred into dialysis tubing (molecular weight cut-off (MWCO) = 3500, Spectrum Lab) filled with TBE. RNA was electro-eluted by placing the montage in TBE for 60 min at 100 V. Eluted RNA were filtered in centrifuge tube (porosity 0.45 µm, VWR) and ethanol precipitated in the presence of 300 mM sodium acetate pH 5.5. After centrifugation and a wash in 70% ethanol, the pellets were dissolved in DEPC-Treated water and quantified with Nanodrop (Thermo Scientific).

In the case of NMM screenings, 2 µL of RNA were incubated with 100 nM of TO1-B in 40 mM Tris-HCl pH 8.0, 50 mM KCl, and 22 mM MgCl2 and TO1-B fluorescence (ex. 492 nm/em. 516 nm) was measured. In the case of NMM resistance measurement, the mixture was further supplemented with 3 µM NMM. In the case of TO3-Biotin screenings, 300 nM of RNA and 100 nM of TO1-B were used with or without 110 nM of TO3-Biotin. Both green (ex. 492 nm/em. 516 nm) and red (ex. 635 nm/em. 665 nm) fluorescence were measured.

**TA cloning and colony screening.** Genes contained in the libraries were diluted in a PCR mixture as immediately above and thermocycled 25 times using a final extension step of 10 min at 72 °C. PCR products were inserted in pTZ57R/T vector following manufacturer's instruction (InsTAclone PCR cloning Kit, Thermo Scientific). Ligation products were recovered by phenol/chloroform extraction and ~100 ng of DNA used to transform Electro-10 blue bacteria (Agilent) placed in a 2 mm electroporation (MicroPulser, Bio-Rad). After an hour of recovery at 37 °C under agitation, bacteria were plated on Luria broth (LB)-ampicillin agar plate and incubated overnight at 37 °C. The colonies were picked, used to inoculate liquid LB and grown at 37 °C until saturation. Plasmids DNA were extracted using 'GeneJet Plasmid Miniprep kit' (Thermo Scientific), and sequences determined by Sanger approach (GATC Biotech).

Single colonies were introduced in 10 µL of a PCR mixture identical to that used for TA cloning and the DNA amplified as above. Two microlitres of PCR product added to 18 µL of IVT mixture containing 2 mM of each NTP, 25 mM MgCl2, 40 mM Tris-HCl pH 8.0, 50 mM KCl, 5 mM DTT, 1 mM Spermidine, 70 µg/mL T7 RNA polymerase and 100 nM TO1-B. The mix was then split in two and one aliquot was supplemented with 3 µM of NMM. The reaction was incubated in a

real-time thermocycler (Mx 3005 P, Agilent) for 2 h at 37 °C and the green fluorescence (ex. 492 nm/em. 516 nm) measured every minute.

**DMS probing of Mangos.** DMS probing consists of four main steps: (i) DMS (denaturing): protocol is adapted from Lorsch and Szostak[36]. RNA of 50 nM was 3′-end-labelled with [32]P pCp and gel-purified. The resulting RNA was incubated in 50 mM HEPES pH 7.5 (volume 50 μL) at room temperature for 30 min. After incubation, 10 μg carrier RNA was added. The sample was then heated to 90 °C for 3 min before the addition of 0.5 μL of 25% DMS (diluted in ethanol) and heated to 80 °C for 1 min, 150 μL ice cold ethanol + 5 μL 3 M NaCl was then immediately added and the sample moved to −20 °C for 30 min. DMS-modified RNA was pelleted by centrifuge at 16,300 RCF at 4 °C for 20 min.

(ii) DMS (native): 50 nM 3′-end-labelled RNA was incubated in 50 mM HEPES pH 7.5, 1 mM MgCl$_2$, 140 mM either KCl or NaCl, with or without 500 nM TO1-B (final volume 50 μL) at room temperature for 30 min. After incubation, 10 μg carrier RNA was added. The sample was then incubated at room temperature for 15 min after the addition of 0.5 μL of 100% DMS. Ice cold ethanol of 150 μL + 5 μL 3 M NaCl was then immediately added and pelleted as for the denaturing DMS protocol.

(iii) Reduction: pellets were re-suspended in 10 μL 1 M Tris buffer pH 8 and 10 μL of freshly prepared 0.2 M sodium borohydride was added. Reaction was carried out on ice and in the dark for 30 min. Reactions were stopped by ethanol precipitation as above.

(iv) Aniline cleavage: to the resulting pellet, 20 μL (1 part Aniline, 7 parts ddH$_2$O, 3 parts glacial acetic acid) were added and incubated at 60 °C for 15 min in the dark. Samples were flash-frozen by placing tubes in liquid nitrogen and lyophilized by speed vacuum centrifuge. Once dry, 20 μL ddH$_2$O was added, the sample was refrozen and lyophilized once again. The pellet was re-suspended in a 50% formamide denaturing solution before being loaded on a 15% polyacrylamide gel (19:1 acrylamide:bis).

**T1 RNase ladder and alkaline hydrolysis ladder.** 3′-end-labelled RNA of 200 pmol was incubated in 20 mM sodium citrate, 6.3 M urea, and 1 U/μL T1 RNase (Thermo Scientific) at 50 °C for 10 min. Samples was flash-frozen in liquid nitrogen for 5 min, heat denatured in denaturing solution at 95 °C for 5 min prior to gel loading. Hydrolysis ladders were generated by incubating in 50 mM NaHCO$_3$ at 90 °C for 20 min and neutralizing using 1 M Tris-HCl.

**Screening for minimal functional Mango motifs.** To identify the minimal functional motif of each Mango, truncated constructs were designed as shown in Supplementary Fig. 7. DNA constructs (IDT) were transcribed by run-off transcription using T7 RNA polymerase. RNA was gel-purified on 10% 19:1 acrylamide:bis polyacrylamide gels. RNA concentrations were determined by NanoDrop readings at A$_{260}$, where extinction coefficients were estimated based on an average 11,000 M$^{-1}$ cm$^{-1}$ per base.

**Affinity measurements of Mango variants.** Fluorescence data were gathered using a Varian Cary Eclipse Spectrometer unless otherwise stated. Fluorescent titrations were measured in a buffer mimicking cellular conditions (WB: 140 mM KCl, 1 mM MgCl$_2$, 10 mM NaH$_2$PO$_4$ pH 7.2, 0.05% Tween-20) to determine binding affinities. Fluorescence was measured at the maximum excitation and emission wavelengths of each complex (Supplementary Fig. 11). Curves were fitted using least squares (Kaleidagraph 4.5) using the following equation for TO1-B:

$$F = F_0 + \frac{F_{max}}{2}\left(K_D + [RNA] + [TO1\text{-}B] - \sqrt{([RNA] - [TO1\text{-}B])^2 + K_D(K_D + 2[RNA] + 2[TO1\text{-}B])}\right),$$

(1)

where $F$ is the fluorescence at a given (RNA), $F_0$ is the unbound fluorescence and $F_{max}$ the maximal complex fluorescence, respectively. When $F_0$ was undetectable, it was set to zero.

Or to the following equation for TO3-Biotin and NMM experiments:

$$F = F_0 + \frac{F_{max}[RNA]}{K_D + [RNA]},$$

(2)

$F_{max}$ was determined using Eq. 1 or Eq. 2, as appropriate.

**Temperature-dependent fluorescence and UV-melting curves.** Temperature-dependence measurements were started at 90 °C decreasing at a rate of 1 °C per min until 20 °C, then returned at 1 °C per min until 90 °C was reached. Fluorescence measurements were obtained at the maximum excitation/emission of the fluorescent complex used and were measured in WB buffer using 1 μM RNA either with or without 5 μM TO1-B. Temperature dependence of fluorescence and absorbance were measured using a Varian Cary Eclipse Fluorescence Spectrophotometer at excitation and emission peaks and a Varian Cary 100 Bio UV-visible spectrophotometer monitoring at 260 nm.

**Circular dichroism.** Circular dichroism spectra were obtained on an Applied Photophysics Chirascan Circular Dichroism Spectrometer using 5 μM RNA, 140 mM monovalent salts and 7 μM TO1-B. Spectra were scanned in 1 nm steps with a bandwidth of 1 nm. Data shown is the average of three repeats. Samples were measured using a 1 mm pathlength quartz cuvette (Starna Cells Inc.).

**Formaldehyde resistance assay.** RNA Mango aptamers were incubated with TO1-B in WB buffer for at least 1 h until equilibrium fluorescence was reached. Formaldehyde was then added such that final concentrations after dilution were 50 nM RNA, 100 nM TO1-B, and 0, 2, 4 or 8% formaldehyde. Fluorescence was measured as a kinetic run at a rate of 2 readings per min using a Varian Cary Eclipse Fluorescence Spectrophotometer, ex/em = 510 ± 2.5/535 ± 5 nm.

**Cell culture and maintenance.** HEK293T cells (293T-ATCC® CRL-3216™) were grown in Dulbecco Modified Eagle's Medium containing 10% foetal bovine serum, 2 mM D-Glucose, 2 mM L-Glutamine, 1 mM sodium pyruvate and 100 U/mL penicillin/streptomycin (Thermo Fisher) and maintained at 37 °C with 5% CO$_2$ in a humidified incubator. Cells used for imaging were cultured in Ibidi glass-bottomed eight-well chamber slides (Ibidi GmbH).

**Plasmid and RNA synthesis.** DNA encoding the F30 folding scaffold[20] was modified to incorporate the Mango RNA sequences (Supplementary Table 5) and ordered from (Integrated DNA Technologies). The DNA was amplified by PCR to incorporate 5′ SalI and 3′ XbaI restriction sites (Supplementary Table 6). PCR products were digested using Fast Digest enzymes (Thermo Fisher) and ligated into SalI/XbaI linearized and shrimp alkaline phosphatase (NEB)-treated pAV5S-F30-2xdBroccoli (Addgene plasmid 66845, a gift from Dr. S. Jaffrey). DNA for both the 5S rRNAs and the scaRNAs were amplified by PCR to incorporate a 5′ BstXI site and a 3′ XhoI site (Supplementary Table 6). PCR products were digested using Fast Digest enzymes (Thermo Fisher) and ligated into BstXI/XhoI linearized and shrimp alkaline phosphatase (NEB) treated pSLQ1651-sgTelomere(F + E) (a gift from Bo Huang and Stanley Qi, Addgene plasmid # 51024). For RNA synthesis, DNA encoding the full 5S-F30-Mango/control sequences were PCR-amplified to include a 5′ T7 RNA polymerase promoter. DNA was transcribed in vitro with T7 RNA polymerase (NEB) at 37 °C for 16 h in 40 mM Tris-HCl, 30 mM MgCl$_2$, 2 mM spermidine, 1 mM dithiothreitol, 5 mM rNTPs, 1 U/μl *Escherichia coli* inorganic pyrophosphatase, 4 U/μl T7 RNA polymerase (pH 7.9). RNA was purified from an 8 M urea, 12% denaturing polyacrylamide gel using 29:1 acrylamide:bis solution (Fisher Scientific). The RNA was excised and eluted in RNA elution buffer (40 mM Tris-HCl pH 8.0, 0.5 M sodium acetate, 0.1 mM EDTA) followed by ethanol precipitation. Fluorescence measurements were taken for each of the RNA constructs using a Varian Cary Eclipse Fluorescence Spectrophotometer (Agilent) containing 40 nM TO1-B, 200 nM RNA, 10 mM sodium phosphate, 100 mM KCl and 1 mM MgCl$_2$ at pH 7.2. Similar measurements were also taken with a limiting amount of RNA (40 nM) in an excess of TO1-B (200 nM) and the results showed a similar trend. U6-Mango/control RNA was synthesized by the PCR amplification of a 5′ T7 sequence to each construct followed by IVT and purification as described above.

**Plasmid and RNA transfection.** RNA was transfected directly into eight-well chamber slides using the lipofectamine-based CRISPRMAX reagent following the manufacturers guidelines (Invitrogen). A final concentration of 62.5 nM RNA in 10 mM sodium phosphate buffer (pH 7.2), 100 mM KCl, and 1 mM MgCl$_2$ was incubated at room temperature with a 1:1 dilution in OptiMEM prior to transfection. The RNA transfected was incubated at 37 °C for 1 h in complete growth medium. FuGene 6 was used to transfect 400 ng of the pSLQ--Mango and control plasmids directly in the eight-well chamber slides following the manufacturer's instructions. Plasmids were left to express between 12–48 h before fixation as described below.

**Cell fixation and immunostaining.** Cells were fixed in PBS containing 4% paraformaldehyde for 10 min on ice followed by permeabilization in 0.2% Triton X-100 for 10 min at room temperature. Cells not requiring immunostaining were washed three times for 5 min each with PKM buffer (10 mM Sodium Phosphate, 100 mM KCl and 1 mM MgCl$_2$) followed by a 10 min incubation in 200 nM TO1-B diluted in PKM buffer before replacing with imaging media (10 mM sodium phosphate, 100 mM KCl and 1 mM MgCl$_2$ 1 μg/mL Hoechst 33258). For immunostaining, cells were first blocked (2% BSA in PBS) for 30 min followed by primary antibody (1:50 − 1:500 dilutions) incubation for 120 min in blocking solution.

Primary antibodies used here were: anti-ribosomal protein S6 (MAB5436, R&D Systems—8 μg/mL), anti-ribosomal protein L7 (ab72550, Abcam—1 μg/mL), anti-ATP5B (ab14730, Abcam—1 μg/mL), anti-GW182 (ab7052, Abcam—5 μg/mL), anti-EEA-1 (ab70521, Abcam—1 μg/mL), anti-LSm3 (NBP2-14206, Novus Biologicals—1 μg/mL), anti-TIAR (sc-398372, Santa Cruz—4 μg/mL). Secondary antibodies used were donkey anti-mouse and donkey anti-rabbit Alexa Fluor 680 (Molecular Probes). Primary antibodies were washed three times for 20 min each in blocking solution followed by incubation with secondary antibody at 1:500 dilution for 60 min, which was subsequently washed as above. After immunostaining the cells were washed and stained in TO1-B and Hoechst 33258 as described previously.

**Fluorescence microscopy and live-cell imaging**. Live and fixed cell images were taken directly in the eight-well chamber slides using a Zeiss Elyra wide-field microscope by exciting at 405 nm (blue), 488 nm (green), 561 nm (red) and 642 nm (far-red) and detecting emission at 420–480 nm, 495-550 nm, 570–640 nm and >650 nm, respectively. Image acquisition for the Mango signal (488 nm laser) used 5 mW of power and 200 ms exposure time, except in the photobleaching-assisted microscopy experiments where 50 ms was used. Due to the observation that the Mango signal is stabilized under pulsed illumination in both fixed and live cells, Z-stacks and time series experiments containing more than one colour were acquired by alternating between each channel for an individual frame, leading to recovery and minimal loss of the Mango signal throughout the acquisition. To visualize the nuclear boundary in live cells, a plasmid expressing a fluorescently tagged histone protein (EBFP2-H2B-6, Addgene plasmid 55243) was transfected using FuGene 6 (Promega) 24 h prior to RNA transfection. RNA was transfected directly into eight-well chamber slides (Ibidi GmbH) as described above, with an additional pre-incubation step with 125 nM of TO1-B prior to the addition of the CRISPRMAX transfection reagent. Upon addition to the imaging chambers, the final effective concentrations of RNA and fluorophore were 10 and 20 nM, respectively. Following incubation of the RNA transfection, the cells were washed once with PBS and replaced with live-cell imaging media (fluorobrite DMEM supplemented with 10% FBS, 2 mM D-Glucose, 2 mM L-Glutamine, 1 mM sodium pyruvate and 10 mM HEPES, Invitrogen). Live cells were maintained at 37 °C with 5% $CO_2$ in a stage top incubator (Tokai Hit).

**Photobleaching-assisted microscopy**. To image the photobleaching of 5S-F30-Mango IV foci as compared with dBroccoli in live cells, 5S-F30-Mango I and IV were directly transfected as described above, whereas 5S-F30-dBroccoli was expressed from pAV5S as previously described[13,20]. Live-cell photobleaching was conducted with a 200 ms frame rate and 5% wide-field laser illumination at 488 nm for 150 frames. To determine the half-life of fluorescence, each photobleaching curve was fit to an exponential decay function. In order to obtain the appropriate signal to noise ratio and time resolution for the analysis of single-step photo-bleaching, 5S-F30-Mango IV foci were imaged in fixed cells with a 50 ms frame rate, 5% wide-field laser illumination at 488 nm for 800 frames. Maximum like-lihood estimation was used to determine each of the photobleaching steps within a trace as previously described[26,27]. The step sizes were subsequently binned and the histogram was fit to a double Gaussian equation.

**Image processing and quantification**. Images were processed using FIJI and spot detection analysis was performed on each maximum projection by the spot detector plugin in the ICY image analysis software, which detects significant foci with a pixel area ≥3 × 3 pixels and intensity ≥300 a.u. A lower threshold of ≥150 a.u. was used to create a population of the apparent background intensities in the control RNA transfections (Fig. 4b). To create the 2D co-localization plots (Supplementary Fig. 16, 17), five to six images for each condition (~100 cells) were processed and the normalized intensity (max = 1, min = 0) of each pixel in both the TO1 channel (y axis) and the AlexaFluor 680 channel (x axis) was plotted. To append a Z-axis density of pixels, Igor Pro was used to carry out a bivariate histogram upon corresponding pixels for both channels and displayed as a heatmap.

**pH titrations**. Mango of 50 nM was incubated with 100 nM TO1-B in the presence of 140 mM KCl, 1 mM $MgCl_2$, and varying pH (50 mM sodium citrate buffer for pH 3–6, 50 mM sodium phosphate for pH 6–8, 50 mM Tris for pH 8–9) for 1 h at room temperature. Mango fluorescence was measured with excitation and emission at 485 nm and 535 nm, respectively.

**In vitro photobleaching measurement**. 5S-F30-Mango I, II, III and IV as well as 5S-F30-Broccoli template regions were placed under the control of T7 RNA polymerase promoter. Genes were PCR-amplified, in vitro-transcribed, purified and quantified as before (see Enrichment measurement section). Then 1 volume of 3 μM RNA solution was added to 1 volume of 2-times concentrated buffer (280 mM KCl, 2 mM $MgCl_2$, 20 mM $NaH_2PO_4$ pH 7.5) supplemented with 3.6 μM of fluorogenic dye (DFHBI-1T for Broccoli aptamer, and TO1-B for Mango apta-mers). The mixture was incubated for an hour at room temperature prior to being loaded into a length of PTFE tubing (Thermo Fisher) and infused into a droplet generator microfluidic device where it was dispersed into 100 pL droplets carried by HFE 7500 fluorinated oil (3 M) supplemented with 3% of a fluorosufactant as described previously[37]. The resulting emulsion was then loaded into 5 μL capillary (Corning) and the montage was imaged on an epifluorescence microscope (TiE, Nikon). Depending on the dye, the emulsion was exposed to a constant illumi-nation wavelength of 470 nm (DFHBI-1T) or 508 nm (TO1-B) at the maximum intensity of the light source (Spectra X, Lumencor), and the emitted fluorescence (respectively, 514 nm ± 24 and 540 nm ± 12) was collected by an Orca-Flash IV camera for 200 ms every 100 ms with ×20 objective (numerical aperture (NA) 0.45). Fluorescence intensity of each picture was then extracted using NiS software (Nikon) and the data were fit to an exponential decay equation to compute the fluorescence half-life.

**Data availability**. The data that support the findings of this study are available from the corresponding authors upon reasonable request.

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

## Acknowledgements

We thank Eric Westhof (Univesité de Strasbourg) and Dipankar Sen (Simon Fraser University) for critical reading of the manuscript, Zewei Ding (Simon Fraser University) for help with plasmid construction, and Julie Biteen (University of Michigan) for help with the step photobleaching analysis. We thank the microscopy core facility at the MRC London Institute of Medical Sciences. We acknowledge support by the MRC core grant (MC-A658-5TY10 to DR) and a NSERC operating grant (P.J.U.). This work has been published under the framework of the LABEX: ANR-10-LABX-0036_NETRNA and benefits from funding from the French National Research Agency as part of the Investments for the future program. It was also supported by the Université de Strasbourg and the Centre National de la Recherche Scientifique (M.R.).

## Author contributions

A.A., S.J., A.C., D.R., M.R. and P.J.U. designed the experiments, analysed the data and wrote the manuscript. A.A. performed the fluorescence activated microfluidic sorting of the R12 library. S.J., A.Ab., A.G. and S.S.S.P. performed subsequent aptamer characterisation and DMS probing. A.C. performed the cellular imaging of the small non-coding RNAs.

## Additional information

**Competing interests:** P.J.U., M.R., S.J., A.A., A.Ab. and S.S.S.P. have filed a provisional patent on aspects of this work. The remaining authors declare no competing financial interests.

