## [Peer Review File · Nature Communications]

REVIEWERS' COMMENTS:

Reviewer #1 (Remarks to the Author):

In this revision, Autour et al expand their previous [REDACTED] manuscript with some live cell work and changes to the text. The authors have largely answered my criticisms of the first iteration of this manuscript and I consider it suitable for publication, subject to correction of the below concerns:

Major:

There simply needs to be some discussion at this point about the kinetics of the thiazole orange-RNA interaction. This dates back to the early days of the studies of this ligand with nucleic acids - and it may not be unique to TO1-Biotin. The high affinity of TO1-Biotin for Mango does not appear to depend on the side chain, yet the side chain makes extensive contacts in the crystal structure. What is going on here? I can appreciate a planned structural biology MS - but this simply must be addressed here. This isn't a first report of an aptamer where this can be glossed over. At this stage, more detail is necessary.

SI Fig 8 - why are higher salt points omitted? What is going on?

SI Fig 18 - where is the comparison to Broccoli and Corn? Please include.

Not happy with the fits in SI fig 19, which skips over numerous points. What is going on here?

Correct "in vivo" to "live cell" throughout. No in vivo work in this paper.

Minor:

TO1-Biotin and TO1 seem to be used interchangeably - clean this up.

I do not like the use of "Spinach-family aptamers" given that significant iteration has been made with these. Something like "HBI-family aptamers" or "GFP-mimic aptamers" would be better.

Reviewer #2 (Remarks to the Author):

This revised manuscript by Autour et al has addressed all of my original concerns. In particular, high-level, intracellular expression of Mango-tagged 5S rRNA show strong nucleolar puncta that is appropriate for 5S localization. The combination of new data, revised text, and rationale presented in the rebuttal letter are convincing. I look forward to seeing this work in print.

Reviewer #3 (Remarks to the Author):

The authors addressed my previous points in full and added impressive new data demonstrating imaging of an RNA fusion construct expressed off a plasmid, a significant advance beyond the transfected aptamer used in the original version of the manuscript. One minor point: On page 16,

the authors state that Mango III and IV have "unprecedented levels of brightness" for a TO1-based system. However, Szent-Gyorgyi et al reported a TO1-protein complex with a quantum yield of 0.47 (Nature Biotech 2007), so the authors should probably revise this sentence to either focus it on nucleic acid-based systems or better yet, mention the Szent-Gyorgyi work to provide a broader context.

Please find below our detailed point by point response to the referee's comments. We wish to thank the reviewers for taking the time to improve our manuscript.

Reviewer #1 (Remarks to the Author):

In this revision, Autour et al expand their previous [REDACTED] manuscript with some live cell work and changes to the text. The authors have largely answered my criticisms of the first iteration of this manuscript and I consider it suitable for publication, subject to correction of the below concerns:

Major:

There simply needs to be some discussion at this point about the kinetics of the thiazole orange-RNA interaction. This dates back to the early days of the studies of this ligand with nucleic acids - and it may not be unique to TO1-Biotin. The high affinity of TO1-Biotin for Mango does not appear to depend on the side chain, yet the side chain makes extensive contacts in the crystal structure. What is going on here? I can appreciate a planned structural biology MS - but this simply must be addressed here. This isn't a first report of an aptamer where this can be glossed over. At this stage, more detail is necessary.

The reviewer raises a very important, but quite complex point: how in detail does the thiazole orange ligand interact kinetically with these high affinity aptamers? The major thrust of our manuscript is that: 1. We have discovered three new and highly fluorescent TO1-Biotin binding aptamers (Mango-II, III and IV) via a new competition-based microfluidic selection. 2. That we can use these aptamers in a live cell context. 3. That some of these aptamers are unusually resistant to formaldehyde and that the aptamers are quite distinct from each other. While in the current work we have characterized many of the properties of these new aptamers we certainly have not characterized all of their commonalities or differences, particularly as it relates to the details of molecular recognition. Therefore, even though important, this characterization is out of the scope of the present study.

We would like to stress that these aptamers are in fact quite different from each other. Mango-I for which we have a published structure (Trachman et al. NCB 2017) does indeed show extensive side chain interactions as noted by the reviewer. In our unpublished work Mango III (and entirely consistent with our characterization presented in the current manuscript), binds TO1-Biotin in a fashion completely distinct from that found in the Mango I structure. Likewise Mango II shows considerable differences in binding mode from that found in Mango I. These differences are fascinating and we feel quite unexpected. We therefore do not feel comfortable discussing these important points in a manuscript whose primary purpose is to introduce these new and functionally diverse aptamers. The space that we have available would not do justice to the topic.

SI Fig 8 - why are higher salt points omitted? What is going on?

A decline in fluorescence was observed for two of the aptamers. This effect was largely limited to the Mango III aptamer (Mango IV shows this effect to a lesser degree as well), which has a distinctly different fold than the other Mango aptamers characterized in this study. This inhibition does not influence our main inference from this data: namely that Mango III responds to low levels of KCl or NaCl much more aggressively than does Mango I, II, and IV, which are much less responsive.

The source of this inhibition has not yet been determined but we speculate that the inhibition is due to misfolding of the aptamer at higher salt levels. Our data strongly suggests that Mango III is a two tiered G-quadruplex structure, while Mango II and IV appear to be three tiered (see Figure 3), just like Mango I (Fig. 1). This difference may account for the observed change in salt dependence but is currently speculative. We added a sentence to the supplemental figure legend 8 to state that we do not fully understand the source of the observed

inhibition.

SI Fig 18 - where is the comparison to Broccoli and Corn? Please include.

The comparison between Mango and Broccoli is already included in Supplementary Figure 14. A comparison to Corn falls outside of the scope of this manuscript because the Corn dye is not commercially available yet. Nonetheless, we believe the manuscript contains ample materials for interested readers to make an informed comparison. This point is also discussed in the discussion section of the manuscript.

Not happy with the fits in SI fig 19, which skips over numerous points. What is going on here?

We are not quite clear which specific fits the reviewer refers to. In panel a, we use a widely accepted maximum likelihood-based algorithm developed in Dr Haw Yang's laboratory (Princeton University) and recently used by, for example, the Dr Julie Biteen's laboratory (U Michigan). In panels b and c, the high intensity points are excluded from the fit because they correspond to two or more fluorophores photobleaching simultaneously, which results in an under determination of the number of fluorophores detected (as seen in the unfitted points observed in c).

Correct "in vivo" to "live cell" throughout. No in vivo work in this paper.

We have corrected this point throughout the manuscript.

Minor:

TO1-Biotin and TO1 seem to be used interchangeably - clean this up.

We have addressed this point by systematically using TO1-B throughout the manuscript and figures.

I do not like the use of "Spinach-family aptamers" given that significant iteration has been made with these. Something like "HBI-family aptamers" or "GFP-mimic aptamers" would be better.

Spinach-family aptamers was replaced with GFP-mimic aptamers as requested.

Reviewer #2 (Remarks to the Author):

This revised manuscript by Autour et al has addressed all of my original concerns. In particular, high-level, intracellular expression of Mango-tagged 5S rRNA show strong nucleolar puncta that is appropriate for 5S localization. The combination of new data, revised text, and rationale presented in the rebuttal letter are convincing. I look forward to seeing this work in print.

We thank this reviewer for his/her comments.

Reviewer #3 (Remarks to the Author):

The authors addressed my previous points in full and added impressive new data demonstrating imaging of an RNA fusion construct expressed off a plasmid, a significant advance beyond the transfected aptamer used in the original version of the manuscript. One minor point: On page 16, the authors state that Mango III and IV have "unprecedented levels of brightness" for a TO1-based system. However, Szent-Gyorgyi et al reported a TO1-protein complex with a quantum yield of 0.47 (Nature Biotech 2007), so the authors should probably revise this sentence to either focus it on nucleic acid-based systems or better yet, mention the Szent-Gyorgyi work to provide a broader context.

We have generalized this section of the text to include the work of Szent-Gyorgyi and we regret not doing this initially. Lines 400-402 now reflect this change.